# Public health benefits of shifting from hospital-focused to ambulatory TB care in Eastern Europe: Optimising TB investments in Belarus, the Republic of Moldova, and Romania

**Sherrie L. Kelly**[1], **Gerard Joseph Abou Jaoude**[2], **Tom Palmer**[2], **Jolene Skordis**[2], **Hassan Haghparast-Bidgoli**[2], **Lara Goscé**[2,3], **Sarah J. Jarvis**[1], **David J. Kedziora**[4], **Romesh Abeysuriya**[1], **Clemens Benedikt**[5], **Nicole Fraser-Hurt**[5], **Zara Shubber**[5], **Nejma Cheikh**[5], **Stela Bivol**[6], **Anna Roberts**[1], **David P. Wilson**[1], **Rowan Martin-Hughes**[1]*

**1** Burnet Institute, Melbourne, Australia, **2** Institute for Global Health, University College London, London, United Kingdom, **3** Department of Infectious Disease Epidemiology, London School of Hygiene and Tropical Medicine, London, United Kingdom, **4** Complex Adaptive Systems Lab, University of Technology Sydney, Sydney, Australia, **5** World Bank Group, Washington, DC, United States of America, **6** World Health Organization, Regional Office for Europe, Copenhagen, Denmark

* rowan.martin-hughes@burnet.edu.au

**Data Availability Statement:** Country-specific data are available in this manuscript and Supplementary

## Abstract

High rates of drug-resistant tuberculosis (DR-TB) continue to threaten public health, especially in Eastern Europe. Costs for treating DR-TB are substantially higher than treating drug-susceptible TB, and higher yet if DR-TB services are delivered in hospital. The WHO recommends that multidrug-resistant (MDR) TB be treated using mainly ambulatory care, shown to have non-inferior health outcomes, however, there has been a delay to transition away from hospital-focused MDR-TB care in certain Eastern European countries. Allocative efficiency analyses were conducted for three countries in Eastern Europe, Belarus, the Republic of Moldova, and Romania, to minimise a combination of TB incidence, prevalence, and mortality by 2035. A primary focus of these studies was to determine the health benefits and financial savings that could be realised if DR-TB service delivery shifted from hospital-focused to ambulatory care. Here we provide a comprehensive assessment of findings from these studies to demonstrate the collective benefit of transitioning from hospital-focused to ambulatory TB care, and to address common regional considerations. We highlight that transitioning from hospital-focused to ambulatory TB care could reduce treatment costs by 20% in Romania, 24% in Moldova, and by as much as 40% in Belarus or almost 35 million US dollars across these three countries by 2035 without affecting quality of care. Improved TB outcomes could be achieved, however, without additional spending by reinvesting these savings in higher-impact TB diagnosis and more efficacious DR-TB treatment regimens. We found commonalities in the large portion of TB cases treated in hospital across these three regional countries, and similar obstacles to transitioning to ambulatory care. National governments in the Eastern European region should examine barriers delaying adoption of

Information, as well as provided in respective country-specific reports as cited in the manuscript and Supplementary Information.

**Funding:** The funding for this work was provided through a trust fund from the Joint United Nations Programme on HIV/AIDS (UNAIDS) (https://www.unaids.org/). The funders had no role in study design, data collection and analysis, decision to publish, or preparation of the manuscript.

**Competing interests:** The authors have declared that no competing interests exist.

ambulatory DR-TB care and consider lost opportunities caused by delays in switching to more efficient treatment modes.

## Introduction

In Eastern Europe, there has been a slow move towards ambulatory care of tuberculosis. This delay may stem partly from legacy financing of TB sanatoriums [1] and bed-based payment modalities, whereby hospitals that refer patients to ambulatory treatment risk budget cuts [2]. Nevertheless, even once effective DR-TB drugs became available, the emergence and persistence of DR-TB has been a consequence of failings in the health care system [3]. With the highest proportions (over 50% of previous treated cases with multidrug-/rifampicin-resistant (MDR/RR) TB) found in several countries in Eastern Europe [4], a legacy of relatively high MDR-TB rates from countries belonging to the former Soviet Union [5], delays in transitioning from hospital-focused to ambulatory TB care will be more costly.

World Health Organization (WHO) guidelines issued in 2011 recommended investment in "systems that primarily employ ambulatory models of care to manage patients with drug-resistant TB over others based mainly on hospitalization" [6]. These guidelines were updated in 2019 [7] and 2020 [8] and maintain the recommendation to treat DR-TB using primarily ambulatory models of care (i.e., services administered in a healthcare facility outside of hospital or in the community including home-based care provided by a community worker). The 2020 WHO guideline update on DR-TB treatment states that "despite the limitations in the data available, there was no evidence that was in conflict with the recommendation, and which indicated that treatment in a hospital-focused model leads to a more favourable treatment outcome" [8]. Moreover, systematic reviews that assessed evidence from a wide range of health settings provided additional support for ambulatory care over hospital-focused models of care for patients infected with multidrug-resistant TB [9–11]. Finally, ambulatory care, which offers patients greater independence during treatment, is also likely to have higher acceptance rates in the longer run. We acknowledge that since randomised clinical trial data are not available, this evidence is primarily based on cohort studies which have inherent selection bias. More affected patients will be clinically eligible to receive hospital-focused care (and will tend to have worse outcomes), and patients less affected will be offered ambulatory care (with better outcomes).

In most Eastern European countries, the TB care model is based on legacy systems of hospital-focused care with injectable DR-TB treatment [12]. Historical models used long-term quarantine and allowed TB patients to recover over time, as these models were developed when effective DR-TB drugs were unavailable and MDR-TB did not yet exist [3].

Regardless of the availability of effective DR-TB drug regimens and updated global health guidance, lengthy hospital-focused care models persist in many Eastern European countries, and barriers to adopting ambulatory DR-TB treatment models still exist. These barriers may involve health financing mechanisms that reimburse based on hospital bed occupancy rates for DR-TB care [2] or financing frameworks based on a restrictive line item budget making purchaser-provider split impossible. To overcome these types of barriers, solutions for health financing reform should consider results-based reimbursement and financing frameworks should allow for a more flexible global budget [3].

Avoiding hospital admissions, particularly to facilities with inadequate mechanisms for infection control, has been a key factor in reducing the risk of nosocomial transmission including the spread of TB and DR-TB [3, 13]. Moreover, with the onset of the COVID-19 pandemic

**Table 1. Key tuberculosis epidemiological, financial, and programmatic information for Belarus, the Republic of Moldova, and Romania.**

| Key category | Belarus [16, 17] | Moldova (Republic of) [18–20] | Romania [21, 22] |
|---|---|---|---|
| Reporting year | 2015 | 2016 | 2018 |
| WHO classification | High MDR-TB burden | High MDR-TB burden | Not high TB burden |
| Est. MDR/RR TB incidence (1000s) | 3.5 (2.8–4.2) | 2.3 (1.9–2.6) | 0.71 (0.56 −0.88) |
| TB financing | | | |
| Total spending | US$50.8 million | US$17.7 million | US$131.5 million |
| Spending for TB treatment | US$47.1 million | US$13.4 million | US$100.6 million |
| *Domestic* | | | |
| % of total | 89% | 77% | 49% |
| Description | Over 75% to hospital care | Not reported | Not reported |
| *International* | | | |
| % of total | 10% | 23% | 11% |
| Description | Nearly 60% to ambulatory care | Not reported | Not reported |
| *Private* | | | |
| % of total | <1% | None reported | 40% |
| Description | Primarily for ambulatory care | Not applicable | Not reported |
| National TB policy | WHO-recommended rapid diagnostic as initial test for all presumed to have TB (92% compliance) and universal access to drug susceptibility testing (100% compliance) [23] | National TB policy for high MDR-TB burden, indicating WRD as the initial diagnostic test for people presumed to have TB [19] | Not available |

Entries were current at the time of each country analysis. MDR/RR = multidrug-/rifampicin-resistant. WRD = WHO-recommended rapid diagnostic.

in early 2020, hospitals became overwhelmed and lockdowns were introduced, and there was an accelerated move to ambulatory TB care to avoid the risk of SARS-CoV-2 infection from seeking TB services in hospital [14]. While some provision for hospital-focused TB care will likely remain in order to deliver specialised care for those with particularly complex cases, the transition to ambulatory care is anticipated to continue.

While the overall burden of TB in Eastern Europe has declined in the last two decades, the incidence of drug-resistant TB has increased. In the three countries examined in this study, Belarus, the Republic of Moldova, and Romania, TB incidence, active TB prevalence, and TB-related deaths declined between the years 2000 and 2015, while the relative share of multidrug-resistant (MDR) and extensively drug-resistant (XDR) TB increased or continued over this period or at least did not decrease in these countries (key country information listed in Table 1). It is worth keeping in mind that improved access to diagnostic technologies and roll-out of rapid molecular diagnostics in high-burden countries has markedly increased the capacity to detect DR-TB over the past decade [15]. Increased diagnosis capacity may contribute to increased reported case numbers for DR-TB.

In this study we collectively examined three investment case study countries in Eastern Europe where TB ambulatory care programs had been defined.

## Materials and methods

### Model and optimisation studies

Mathematical optimisation of TB spending was conducted using Optima TB, a dynamic population-based model of TB transmission. The model, which is fully described by Goscé et al.

**Table 2. Ambulatory-focused treatment modalities considered in the modelling studies for Belarus, the Republic of Moldova, and Romania.**

| Ambulatory-focused TB treatment regimens | Belarus (2015) [16] | Republic of Moldova (2016) [18] | Romania (2018) [21] |
|---|---|---|---|
| DS[1] | Standard and incentivised[2] modalities | Standard modality | Standard and DOTS modalities |
| MDR short-course[3] | Standard and incentivised[2] modalities | Not considered[4] | Standardised under DOTS and supportive observation |
| MDR long-course[5] | Standard and incentivised ambulatory[2] modalities | MDR standard and MDR plus[6] | Standard with and without Bedaquiline or delamanid |
| **XDR** | Standard and incentivised ambulatory[2] modalities | Pre-XDR and XDR standard and with the addition of new drugs | Standard without Bedaquiline or delamanid |
| **New and repurposed XDR drugs** | Incentivised ambulatory[2] modalities include the addition of Bedaquiline, clofazimine, and linezolid | New drugs included Bedaquiline, linezolid, imipenem/cilastatin, and amoxicillin/clavulanic acid | Standard with the addition of Bedaquiline or delamanid |

Years reported below each country column header indicate the reporting year.

[1]The WHO recommends ambulatory care of drug-susceptible (DS) TB with daily six-month rifampicin-based regimen initiation phase of two months of isoniazid (H), rifampicin (R), pyrazinamide (Z) and ethambutol (E), followed by a continuation phase of four months of HR (2HRZE/4HR).

[2]Incentivised ambulatory indicates that financial incentives were incorporated.

[3]MDR short-course is the shorter MDR regimen with a four month intensive phase (extended to six months in case of delayed sputum smear conversion) containing high-dose gatifloxacin or moxifloxacin, kanamycin, prothionamide, clofazimine, high-dose isoniazid, pyrazinamide and ethambutol followed by a continuation phase of five months containing gatifloxacin or moxifloxacin, clofazimine, ethambutol and pyrazinamide.

[4]Not considered indicates the regimen was not being considered for implementation at the time the country study was conducted.

[5]MDR long-course is the longer MDR regimen with treatment up to 18 to 24 months.

[6]MDR plus represents treatment with re-purposed medications such as Linezolid (LNZ), imipenem /cilastatin, and Amoxicillin/Clavulanic acid.

DOTS = directly observed treatment, short-course. DS = drug-susceptible. MDR = multidrug-resistant. XDR = extensively drug-resistant.

[24], was developed to inform evidence-based priority setting processes for strategic TB programmatic planning and resource prioritisation, and has been validated across several settings. Studies were conducted for three countries in Eastern Europe: Belarus, the Republic of Moldova, and Romania. An analysis was conducted in 2016–2017 for Belarus with a full description of the methodology provided in the published country report [16]. Analyses were conducted in 2017–2018 for the Republic of Moldova as described in [18] and in Romania from 2018–2019 as described in [21]. The objective of these published studies was to identify the most cost-effective resource allocation across existing and prospective TB diagnosis and treatment modalities to minimise a combination of active TB cases, prevalence of active TB, and TB-related deaths by 2035. A primary focus, however, was to determine the health benefits and cost savings that could be gained by shifting from hospital-focused to ambulatory TB care. This approach aligns with targets established in the National TB Programme strategic plans for the countries considered.

## TB treatment modalities

Table 2 lists the ambulatory-focused interventions that were implemented in each country at the time each country study was conducted. Duration of hospital-focused and ambulatory TB treatment by modality for each country are shown in Table A in S1 Text for Belarus, Table C in S1 Text for Moldova, and Table D in S1 Text for Romania.

## Study data and costing

For each country study, epidemiological, program, and cost data were collated as part of investment case studies that were conducted in collaboration with in-country experts as detailed in country reports [16, 18, 21]. As part of remote working sessions and in-country

workshops, data were compiled for each country in data entry spreadsheets and in the Optima TB modelling tool as part of the Optima TB modelling process as described by Goscé et al. [24]. Literature reviews were also conducted to inform model parameters as and to support assumptions where necessary as described in each country report [16, 18, 21].

Data were collated for population definitions and size, notified TB cases, additional epidemiological metrics, comorbidity, testing and treatment, and TB programs (as detailed in Tables 2.1 of the country reports [16, 18, 21]). Common sources for these data include the UN World Population Prospects [25] and WHO TB Epidemic Profiles with key epidemiological estimates, and values for the status of drug resistance and key treatment indicators provided in Sections 3 of the country reports for Belarus [16], Moldova [18], and Romania [21]. Model inputs were reviewed and finalised together with the in-country teams. Optima TB data spreadsheets and projects for each country, Belarus, Moldova, and Romania, can be made available upon request and with country permission.

TB costing exercises were carried out for each country as part of the original investment case studies. Costs represent the full cost of delivering a given intervention including commodities, delivery costs, staff time and TB-related costs outside the TB programme, such as TB-related hospitalisation by treatment modality and facility costs. Costs were determined through bottom-up costing capturing health system costs and not additional societal costs. As ambulatory care permits reduced societal costs (e.g. days of lost productivity), and while societal costs were not captured here, they would only strengthen these findings. Costing calculations were carried out by country teams for the most recent two to three full years of available cost data for each country. Discounting was not applied. Costing data are provided in Table B in S1 Text for Belarus, Table C in S1 Text for Moldova, and Table E in S1 Text for Romania.

For Belarus, baseline spending by TB intervention and treatment type was established using the 2015 expenditures from WHO national health sub-accounts. These were triangulated with unit costs from other countries and international costing data to establish estimated spending by intervention as shown in the Table B in S1 Text. TB drug cost per course of treatment by modality include domestic and international donor funding (the Global Fund) for 2015.

For Moldova, 2016 expenditure data sourced from WHO databases and reports, national TB reports for the WHO and Ministry of Health, and National TB Programme records were triangulated with other unit cost data to establish estimated spending by intervention as shown in Table C in S1 Text. Costs were calculated considering the number registered TB patients, annualised costs, with other costs accounting for adverse drug reaction monitoring costs including costs of tests (such as audiometry, thyroid function, liver functioning, and electrocardiogram), which were mainly associated with drug resistant cases of TB.

For Romania, bottom-up costing calculations for all treatment programs were carried out using average daily costs from hospital data. An average cost per ambulatory interaction was also derived and applied to screening programs and ambulatory treatment following the initial hospitalisation period as shown in Table E in S1 Text.

Spending was reported in 2015 USD for Belarus [16], in 2016 USD for Moldova [18], and reported in Euros for Romania [21] converted here to 2016 USD corresponding to the year spent (at the time of original analyses).

## Model calibration and cost-functions

Country models were calibrated primarily to TB case notifications and registered TB deaths. Cost-functions representing the relationship between spending and coverage, and coverage and outcome were generated for each intervention. Calibrations and cost functions were validated together with in-country stakeholders.

## Optimisation approach

Using each country model, allocative efficiency projections were simulated for the total TB budget including for prevention, diagnostic, and treatment interventions. The potential for expanded diagnosis through active case finding was informed by country stakeholders when setting the model constraints for each program, and it is assumed that all those diagnosed will be eligible to receive TB treatment. Optimisation solutions for each country that best met the defined objectives were selected. From these reallocations, optimised TB treatment program spending for hospital-focused and ambulatory care, as well as for other treatment interventions (palliative care, prison-based treatment) were compared with the latest reported treatment spending. As part of the total TB budget optimisation, if less expensive but equally effective ambulatory TB treatment interventions (with costs provided in Table B in S1 Text for Belarus, Table C in S1 Text for Moldova, and Table E in S1 Text for Romania) are determined to be more impactful in achieving defined objectives by 2035 than hospital-focused treatment, then the model algorithm will relocate resources accordingly. Treatment modalities include an average number of days of hospital-focused care that accounts for some portion of DR-TB patients who still require hospital-focused DR-TB care due to the severity of their illness or comorbidity status. Additionally, each country applied different maximum constraints on the proportion of people with TB who would be eligible for a shift to ambulatory care. In Belarus, a constraint was applied such that at least 30% of the most recently reported spending on hospital-focused treatment for all TB strains would continue, as well as 20% of involuntary isolation. In Moldova, the constraint was at least 10% for DS-TB and MDR-TB and 15% for XDR-TB. In Romania, no constraint was used for hospital-focused care but a 40% minimum-constraint was applied for DOTS and a 50% minimum-constraint for new drug regimens (MDR and XDR-TB).

For these modelling analyses it was assumed that any savings from prioritising more cost-effective ambulatory TB services would be reinvested in TB programs, versus disbursed, at least in part, to other health areas. However, as reported at the time of the original analyses and explored through follow-up interviews with study country teams (conducted in 2021), there have been structural limitations with healthcare financing that have restricted opportunities to reinvest savings from one area of TB programming into another. These limitations should be examined. Nevertheless, whether governments decide to reinvest savings directly in TB programmes, in other areas of health, or in non-health related sectors, there are opportunities for the country to benefit. Therefore, any potential gains should be pursued and lost opportunities avoided.

## Outcomes

As part of optimising resource allocations for Belarus, Moldova, and Romania less costly but equally effective ambulatory DR-TB treatment modalities would be prioritised over hospital-focused care for each country model. These prioritisations will lead to cost savings, which, in the models, would be reinvested in higher-impact interventions including early TB diagnosis, thus allowing additional increases in ambulatory treatment coverage. Financial gains including the percentages and amount of cost savings for each country and collectively from shifting to ambulatory DR-TB care were reported and calculated, respectively. If DR-TB treatment was shifted to ambulatory care, the public health impact outcomes were estimated including the annual percentages and numbers of additional people that could be treated, the numbers of cumulative new active TB cases and TB-related deaths that could be averted by 2035, and the projected reductions in active TB prevalence that could be achieved over this period. This analysis draws together common results and conclusions from TB budget impact studies for

Belarus, Moldova, and Romania, with a focus on projected health benefits and financial gains that could be realised by prioritising ambulatory DR-TB care.

## Results

Moving from hospital-focused to ambulatory TB care would yield positive public health benefits in all three country settings.

### Optimised TB budget reallocation for Belarus

For Belarus, transitioning from the 2015 model of hospital-focused TB care to ambulatory care could reduce TB treatment costs by nearly US$15 million or 31% by 2035 (Fig 1). At the time of this analysis, it was projected that TB cases in Belarus would decline in the future, which would result in fewer people needing TB treatment. Immediate savings from transitioning from involuntary isolation and other hospital-focused treatment to ambulatory care, as well as savings if new cases decline as projected meaning reduced need for treatment, should be reallocated to higher impact program interventions and delivery solutions. These include providing incentives to improve patient adherence and ambulatory care outreach, procuring new, more efficacious drug regimens for MDR and XDR-TB, scaling up rapid molecular diagnostics, enhancing active case finding among high-risk populations, and enhancing contact tracing.

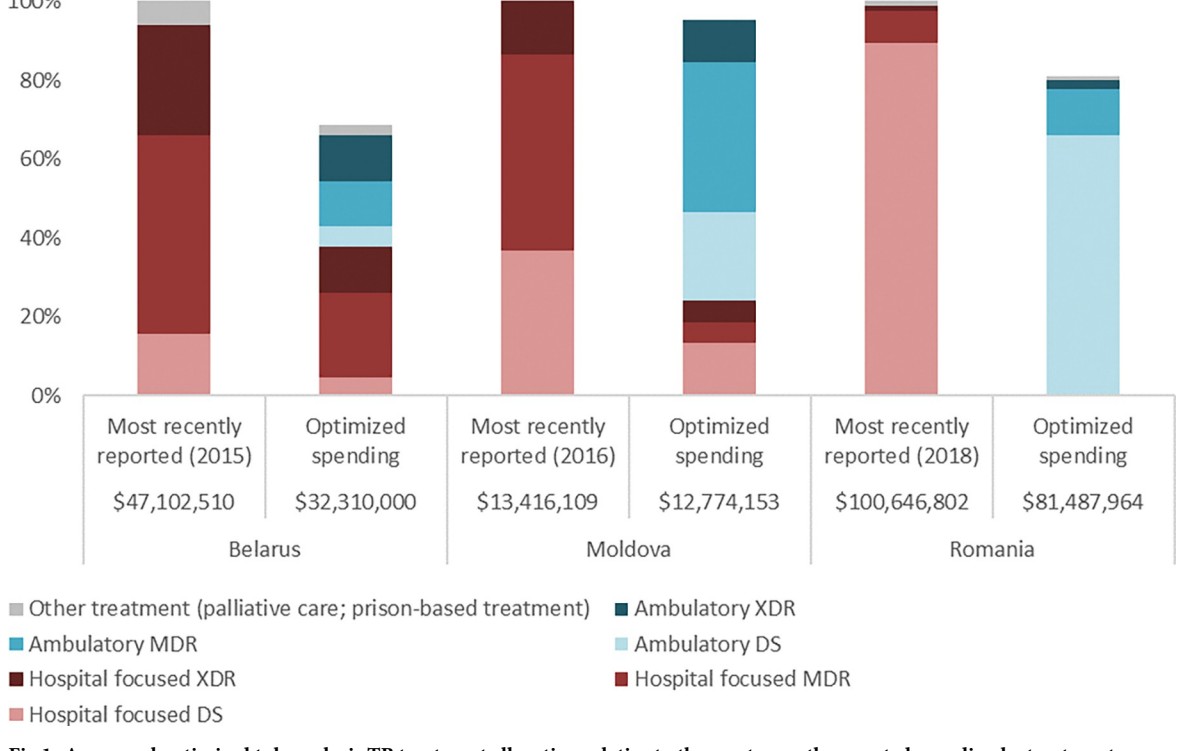

**Fig 1. An annual optimised tuberculosis TB treatment allocation relative to the most recently reported spending by treatment modality (for the given reporting year) represented as a percentage of total TB programme spending for Belarus [16], Moldova [18], and Romania [21].** TB treatment interventions include hospital-focused and ambulatory care for DS, MDR, and XDR-TB, as well as other treatment interventions (palliative care, prison-based treatment). Values for the most recently reported annual TB treatment budget are shown adjacent to an annual optimised resource allocation of TB treatment interventions for each country. Spending is reported in 2015 USD for Belarus, and 2016 USD for Moldova and Romania. DS = drug-susceptible. MDR = multidrug-resistant. XDR = extensively drug-resistant.

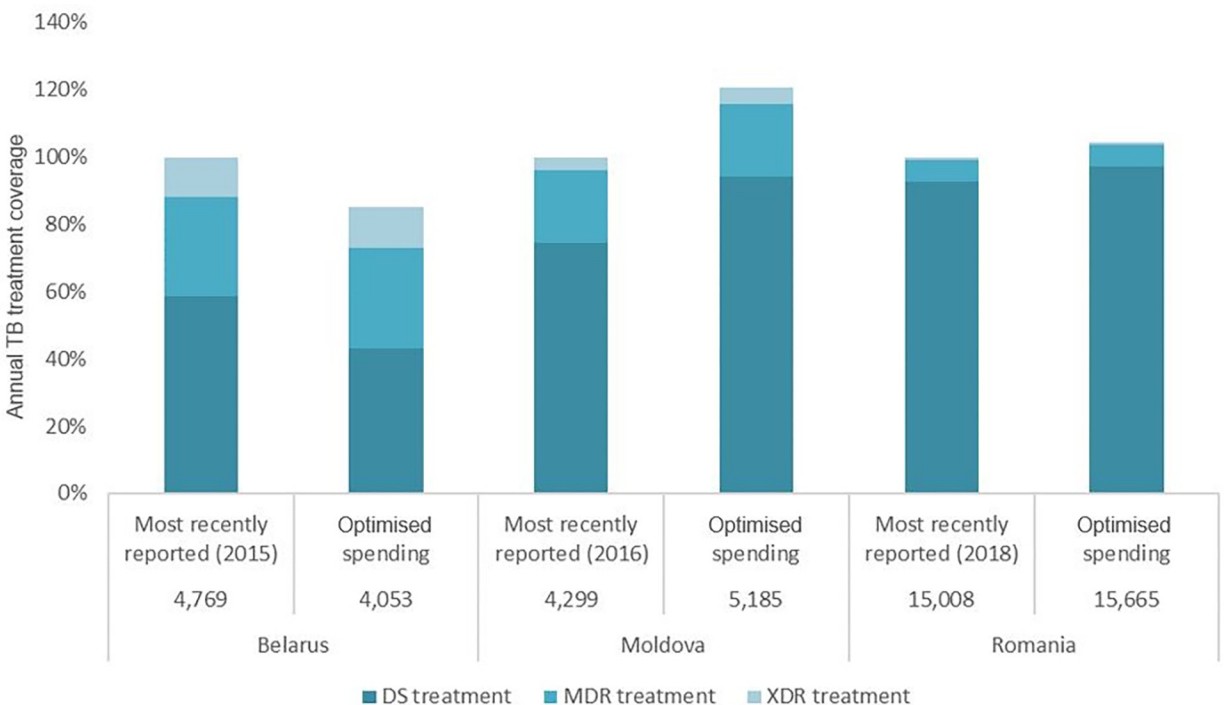

**Fig 2. Annual TB treatment coverage under optimised allocation of resources for all TB interventions compared with the most recently reported treatment coverage for Belarus [16], Moldova [18], and Romania [21].** Most recently reported annual percentage of people on TB treatment (with reporting years shown below the respective bars) are shown alongside annual percentage of people on TB treatment projected under optimised allocation for each country. Total numbers of people covered are indicated below the respective bars. DS = drug-susceptible. MDR = multidrug-resistant. XDR = extensively drug-resistant.

For example, in Belarus, there were 264 patients treated in hospital for DR-TB in 2015. These modalities had the highest unit costs, $21,482 for a full long-course of MDR treatment and $28,840 for XDR treatment in 2015 USD. US$16.6 million was spent on these modalities accounting for 26.8% of all TB-related spending in that year. As well, this transition to ambulatory care resulted in reductions in duration of hospital stay from 60 to 14 days for drug susceptible (DS) TB treatment, 210 for MDR to 45 days for long-regimen and 30 for short-regimen, and 270 to 60 for XDR-TB care.

Fig 2 shows the most recently reported TB treatment coverage values alongside the projected annual TB treatment coverage values (the same for each year from 2015 to 2035) for the same total amount of national TB budgets for Belarus, Moldova, and Romania optimised to best achieve objective targets through to 2035.

Coverage values by type of TB treatment (DS, MDR, and XDR) are also represented graphically as a percentage of treatment need. Total numbers of people covered are shown below the respective bars. However, since Belarus is the only country considered where TB cases are projected to decline – meaning less people would need treatment – over 30% fewer people will need DS treatment under optimised allocation. It is projected that cost-effective reallocation will lead to marginal coverage increases of drug-resistant TB modalities in Belarus.

## Optimised TB budget reallocation for Moldova

As part of the modelling analysis conducted for Moldova, it was estimated that prioritising ambulatory care could reduce treatment costs by an estimated 5% for any given year from 2015 to 2035, potentially freeing up approximately US$0.6 million for reallocation to higher

impact interventions including reinvestment to increase treatment coverage. The largest relative proportion of this saving comes from MDR and XDR-TB treatment programs that have the longest duration of treatment programs at a duration of 18 to 24 months. Lengthy hospitalisation is the primary cost driver of the TB response in Moldova. Based on national program records, the duration of hospitalisation could be reduced substantially from 40 to 14 days on average for DS-TB treatment to align with international best practice. Hospitalisation for drug-resistant TB treatment could be reduced from 45 days for long-course MDR-TB and to 30 days for short-course, and from 127–195 days to 60 days for XDR-TB [18]. Reduced hospitalisation for XDR-TB cases (excluding pre-XDR) would allow for increasing coverage by up to 153%, which would in principle allow nearly every person with XDR-TB who is aware of their status to be on treatment with new Bedaquiline-based pre-XDR and XDR regimens where eligible or standard regimens where not available. It was recommended that any resources freed up by changing treatment modalities should be invested in higher impact interventions and more efficacious treatment regimens. These include provision of incentives for providers of ambulatory TB care, procurement of new, more efficacious drug regimens for MDR-TB and XDR-TB, scale up of rapid molecular diagnostics, enhanced active case finding among high-risk populations, and enhanced contact tracing [18].

### Optimised TB budget reallocation for Romania

Finally, the analysis for Romania also confirmed that transitioning to ambulatory care after a reduced initial hospitalisation could reduce the cost of TB treatment by US$19.2 million by 2035, a 19% reduction in current expenditure. Reductions in duration were as follows, from 67 to 21 days for DS-TB, from 180 days to 30–60 days for MDR-TB, and from 270 days to 120–180 days for XDR-TB, including the use of short directly observed treatment, short course (DOTS) where appropriate [21]. Under optimised allocation, treatment coverage in Moldova and Romania would surpass the most recently reported (at the time of analysis) need, 121% and 104%, respectively.

### Optimised TB budget reallocation for Belarus, Moldova, and Romania

Taken together, if resources for TB were optimally reallocated from 2015 to 2035 for Belarus, Moldova, and Romania, including prioritising less expensive but equally effective ambulatory TB care (therefore more cost-effective) over hospital-focused care, and assuming these savings remained in the TB budget and were optimally reinvested across TB interventions, then new active TB infections could be reduced by 9% in Moldova (1% in Romania and 7% in Belarus), active TB prevalence per 100,000 reduced by 44% in Moldova (5% in Belarus and 27% in Romania), and TB-related deaths reduced by 48% in Moldova (5% in Belarus and 21% in Romania) over this period (Fig 3 with corresponding estimates reported in Tables F–H in S1 Text). Focusing on maximising TB outcomes, not considering potential benefits for other areas of health, modelling shows these savings should be optimally reinvested in TB prevention, diagnosis, and ambulatory treatment interventions to increase treatment coverage. In each country, increased investment in active case finding (particularly in high incidence areas and to target high-risk groups) and prevention was projected to lead to rapid decreases in the prevalence of active TB prevalence and TB-related mortality, but the high burden of latent TB means that new active TB infections are projected to decline more slowly.

### Discussion

Importantly, evidence suggests that ambulatory care for those with drug-resistant TB infection has at least the same treatment outcome as hospital-focused care [8]. Moreover, Williams and

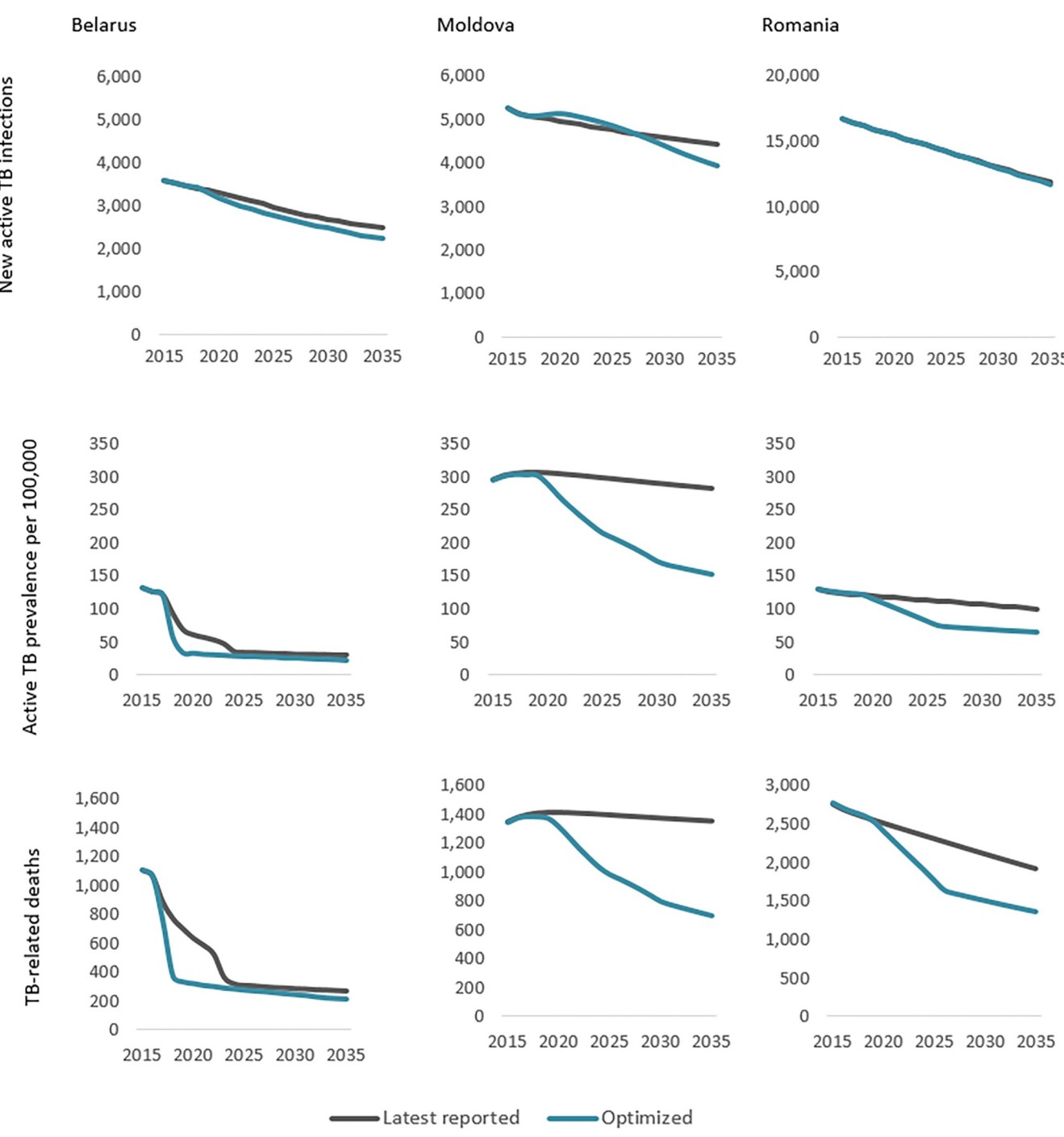

**Fig 3. Projected reductions in new active TB infections, active TB prevalence, and TB-related deaths under optimal allocation of treatment resources from 2015 to 2035 for Belarus [16], Moldova [18], and Romania [21].** This includes prioritisation of less expensive ambulatory care and resulting savings being optimally reinvested in TB prevention, diagnosis, and additional ambulatory treatment.

colleagues observed better MDR-TB treatment success for ambulatory treatment compared with traditional hospitalisation for nine countries in Africa, Asia, and Eastern Europe [26], as was also reported for the Republic of Macedonia [27]. Similarly, Ho and colleagues reported

that success was more likely for ambulatory care from eight studies in Africa, Asia, and the USA [9]. The 2019 WHO guidelines on DR-TB conditionally recommend that "patients with MDR-TB should be treated using mainly ambulatory care rather than models of care based principally on hospitalization" [7]. Despite low quality evidence from observational studies used to inform the 2020 updated WHO guidelines on DR-TB care, the guidelines state that "there was no evidence that was in conflict with the recommendation, and which indicated that treatment in a hospital-focused model of care leads to a more favourable treatment outcome" [8]. Here we demonstrate that transitioning from hospital-focused to ambulatory DR-TB treatment could yield savings of 31%, 5%, and 19% in Belarus, Moldova, and Romania, respectively, while achieving at least comparable projected treatment outcomes (Figs 2 and 3). It is recommended that these savings be optimally reinvested in TB prevention, diagnosis, and ambulatory treatment to achieve increased treatment coverage and further health gains.

This cross-country examination facilitates the comparison of cost per treatment course by TB treatment modality (DS, MDR, and XDR) either delivered in hospital or through ambulatory care. This revealed the potential savings that could be realised by prioritising ambulatory care. However, prioritising TB treatment delivery from hospital-focused to ambulatory care [28–30] will involve more than decision-making on funding reallocation. This transition will require shifting emphases in care models through changes in clinical guidelines, changes in how funding flows to facilities, or through incentives. This may also include task shifting and other changes to human resourcing, as well as changes in demand-side expectations for hospital versus ambulatory care. Lastly, in many settings TB care financing reform may not be a short-term process and may require different approaches and timeframes.

These original three country studies examine a reduction in unnecessary hospitalisation in-line with global TB care guidelines [7, 8], but do not remove the need for hospitalisation entirely. For the majority of cases there is no clinical benefit of DR-TB treatment being administered in hospital. Hospitalisation is clinically indicated only for a relatively small number of cases. The motivation to transition from hospital-focused to ambulatory DR-TB care is to not only to save costs for the health system and for patients [31] and their families (including lost income due to hospital stays estimated at 60% of out-of-pocket expenses as reported in a 2014 review in low- and middle-income countries [24]), but also to reduce the risk of nosocomial transmission [13].

From 2014 to 2018, 14 of the 15 countries in Eastern Europe and Central Asia (EECA) reduced their number of bed days per TB patient. Overall Belarus was able to reduce their overall bed days for treatment by over 20% from 2015 to 2018. The number of hospital bed days per MDR or XDR patient per year were reduced from 120 to 115 days, although this is largely in line with the reduction in the number of TB cases that were projected. Romania was able to reduce their bed days per patient by 11% over this period with the relative size of the reduction influenced by both the percentage of TB patients hospitalised and the average length of stay if hospitalised [22].

The modelling study in Belarus provided evidence that led to a recommendation to strengthen ambulatory care through incentives to improve health worker outreach support and patient adherence. It was suggested that this recommendation be fulfilled using a combination of delivery solutions, which are likely to improve treatment outcomes. It is acknowledged that enhanced ambulatory care requires a reform of tuberculosis care financing to replace bed-based payment with outcomes-based financing. In Moldova, as reported in the 2020 WHO Global TB report, "It is also evident that some EECA countries have markedly reduced their use of hospitalisation and have changed their model of care for people with drug-susceptible TB". As noted previously, from 2014 to 2018, 14 of the 15 EECA countries reduced the number of bed days per person [22]. The size of the reduction, which is influenced

by the percentage of people with drug-susceptible TB who are hospitalised and the average length of stay if hospitalised, ranged from 21% in the Republic of Moldova to 81% in the Russian Federation. As such, new active case-finding modalities were being introduced as of 2019. Mobile outreach vans were being piloted to target high-risk populations, with the aim of ensuring early diagnosis and treatment for people who are typically hard to reach.

Here we provide a comprehensive cross-country assessment of findings from the three TB investment case studies that were conducted independently in Belarus, Moldova, and Romania to assess the benefit of transitioning from hospital-focused to ambulatory TB care across multiple countries within the EECA region. This cross-country comparison revealed that there are commonalities in the large portion of TB cases treated in hospital-focused care across the region, and similar obstacles to transitioning to ambulatory care. The recommendations from these three countries are regionally relevant, and provide for more robust evidence to encourage national governments within the region to expedite the examination of barriers delaying the adoption of ambulatory DR-TB care towards more efficient and effective treatment modes. Similarly, another study in Bulgaria showed that while it would not affect TB burden, switching to ambulatory care for eligible TB patients would be markedly cost-saving [32].

A separate study is underway together with national stakeholders to assess whether recommendations from these modeling studies have been adopted, how they have been implemented, and what benefits may have been gained as a result, as well as lessons learnt. This new study will include these countries in Eastern Europe, but other country studies and disease areas will also be included.

The COVID-19 pandemic has resulted in a shift to ambulatory care to avoid the risk of SARS-CoV-2 infection [14]. This was achieved through technical advances including telehealth, video supported treatment, and other lower contact service delivery approaches. Many of these innovations were in place before the pandemic, but the pandemic prompted the transition to utilise these modalities making it more convenient and decreasing the burden for both patients and providers in ambulatory settings. It is anticipated that many of these care options will continue, even once the need for the COVID-19 response is lessens. Given the potential gains from furthering shift towards ambulatory care, as estimated here, it would be advantageous for TB programme planners to continue incorporating this shift in service delivery into ongoing TB response plans.

Following global guidelines to transition away from hospital-focused to ambulatory DR-TB care [8] there are other benefits beyond cost savings, which were not captured in this analysis. Other benefits include reduced nosocomial transmission-related health systems costs [13], cost (direct and indirect) to the patient [31]. It may also be worth exploring the cost-effectiveness of integrating DR-TB care services with other health programs, particularly those delivered more readily in ambulatory care settings, such as mental health services and alcohol cessation support. One such example is for people coinfected with TB and HIV; co-treatment could be decentralised through ambulatory care and therefore be more patient-centred, could result in healthcare cost savings, reduced loss in income through avoided hospital stays, and other benefits [33].

An international systematic review of the evidence supports the assumption that ambulatory care could achieve current coverage levels in target populations [11]. A meta-analysis of 540 articles reported no statistical difference for treatment outcome rates (success, death, default, and failure), between ambulatory and hospital-focused delivery of TB care. The review found that standard ambulatory care can be as effective as hospital-focused care [11]. There is also evidence to suggest that ambulatory care that is enhanced by specific incentives might be more effective than standard ambulatory care. A Cochrane review suggested that ambulatory care coupled with cash incentives for patients may be more effective than non-incentivised ambulatory care, particularly among high-risk groups [34]. A WHO review of evidence also

suggests improvements in treatment adherence through food and financial support as well as TB care enhanced through a mix of interventions [35]. Considerations around a complete shift from hospital-focused to ambulatory care are that comorbidities, including alcohol use disorder, and coinfection with HIV (non-homogeneous), are also common in this region. In future, more complex cases will likely still need at least some hospitalised care.

As with any modelling study, there are limitations to our approach. First, and mainly, these are related to limited individual-level detail typical of population-level models to capture the homogeneity between individuals belonging to the same population groups. Second, model parameters with missing data were handled by triangulating values from the literature and those provided in consultation with country experts. To note, any data paucity on the progression of infection was largely obviated through model calibration. Third, determining an optimised resource allocation depends on the availability of estimated effectiveness for each intervention. Effectiveness parameters were informed either using setting-specific data, where available, or using global literature values. However, non-setting specific effectiveness values can vary across settings and may not be contextually representative. Fourth, annual optimised resource allocations are projected to change immediately and remain the same year-on-year throughout the simulation period. However, dynamic future programmatic, service delivery, and intervention costing changes or disruptions, such as those caused by the COVID-19 pandemic, are not reflected in these results. Fifth, as part of reinvesting any savings from transitioning to ambulatory DR-TB care in order to increase treatment coverage, options for increasing treatment adherence such as abbreviated treatment regimens, expanded patient incentives, and community support interventions should be explored and benefits tracked to inform future analyses. While these reinvestment options were assessed for individual country studies in response to policy questions, we cannot compare prioritisation of these enhanced treatment modalities across the studies or draw absolute regional conclusions. Last, while we acknowledge that standard DR-TB drug regimens and associated costs have changed since these three original studies were completed, the need to transition those eligible from hospital-focused to ambulatory care, as recommended here, remains relevant.

As shown in this study, replacing hospital-focused care administering injectable DR-TB treatment with ambulatory care administering oral regimens for drug-susceptible and drug-resistant TB, which have fewer side effects and favour decentralised TB care models [7, 8] is both cost-saving and prevents lengthy stays in hospital. Countries are encouraged to adopt ambulatory care models, where clinically indicated. As a follow-on to these studies, most countries in Eastern Europe are currently transitioning towards ambulatory TB care [36].

## Supporting information

**S1 Text.**
(DOCX)

## Acknowledgments

The authors gratefully acknowledge contributors to the country studies that formed the basis of this work, including the following representatives: Dzmitry Klimuk, Alena Skrahina, Henadz Hurevich (Republican Scientific and Practice Centre for Pulmonology and Tuberculosis, Belarus); Inna Nekrasova, Marina Sachek, Vassily Akulov (Republican Scientific and Practice Centre for Medical Technologies, Belarus); Alena Tkatcheva (Ministry of Health, Belarus); Dragutan Cristina, Lilia Gantea (Ministry of Health, Labour, Social Protection, Moldova); Rita Seicas (Center for Health Policies and Studies, Moldova); Victoria Petrica (Project

Coordination and Implementation Unit, Moldova); Valeriu Sava (SDC, Moldova); Sofia Alexandru, Diana Condratchi, Andrei Corlateanu, Valeriu Crudu, Nicolae Nalivaico, Valentina Vilc, Liudmila Marandici ('Chiril Draganiuc' Institute of Phthisiopneumology/National Tuberculosis Programme, Moldova); Munteanu Ioana (National Tuberculosis Programme, Romania); Moldova Adriana Socaci, Beatrice Mahler-Boca, Domnica Ioana Chiotan, Gilda Popescu, Mihaela Stefan Nicoleta Cioran (Romania National Institute of Pulmonology); Amalia Serban, Ana-Maria Ciobanu, Costin Iliuta, Mihaela Bardos (Ministry of Health, Romania); Dana Farcasanu, Daniel Ciurea (Center for Health Policies and Services, Romania); Fidelie Kalambayi (Romanian Angel Appeal); Mihnea Dosius (Romania National School of Public Health); Viatcheslav Grankov Valentin Rusovich, Andrew Siroka, Cassandra Butu (World Health Organization); David Kokiashvili George Sakvarelidze, Sandra Irbe (Global Fund); Azfar Hussain, Janka Petravic, Cliff Kerr (Burnet Institute); Ibrahim Abubakar, Marius Nasta (University College London); and Feng Zhao, Marelize Görgens, Irina Oleinik, Irina Guban, Hanna Shvanok, Cristina Petcu, Huihui Wang, Jaime Nicolas Bayona Garcia, David Wilson (World Bank).

## Author Contributions

**Conceptualization:** Sherrie L. Kelly, Clemens Benedikt, Nicole Fraser-Hurt, Zara Shubber, Nejma Cheikh, David P. Wilson, Rowan Martin-Hughes.

**Data curation:** Sherrie L. Kelly, Gerard Joseph Abou Jaoude, Tom Palmer, Rowan Martin-Hughes.

**Formal analysis:** Sherrie L. Kelly, Gerard Joseph Abou Jaoude, Tom Palmer, Sarah J. Jarvis, Rowan Martin-Hughes.

**Funding acquisition:** David P. Wilson.

**Investigation:** Sherrie L. Kelly, Gerard Joseph Abou Jaoude, Tom Palmer, Sarah J. Jarvis, Clemens Benedikt, Zara Shubber, Rowan Martin-Hughes.

**Methodology:** Sherrie L. Kelly, Sarah J. Jarvis, David J. Kedziora, Romesh Abeysuriya, David P. Wilson, Rowan Martin-Hughes.

**Project administration:** Sherrie L. Kelly, Anna Roberts, Rowan Martin-Hughes.

**Software:** Lara Goscé, Sarah J. Jarvis, David J. Kedziora, Romesh Abeysuriya, David P. Wilson, Rowan Martin-Hughes.

**Supervision:** Sherrie L. Kelly, Jolene Skordis, David P. Wilson.

**Validation:** Sherrie L. Kelly, Gerard Joseph Abou Jaoude, Tom Palmer, Jolene Skordis, Nicole Fraser-Hurt, Stela Bivol, David P. Wilson, Rowan Martin-Hughes.

**Visualization:** Sherrie L. Kelly, Rowan Martin-Hughes.

**Writing – original draft:** Sherrie L. Kelly, Rowan Martin-Hughes.

**Writing – review & editing:** Sherrie L. Kelly, Gerard Joseph Abou Jaoude, Tom Palmer, Jolene Skordis, Hassan Haghparast-Bidgoli, Lara Goscé, Sarah J. Jarvis, David J. Kedziora, Romesh Abeysuriya, Clemens Benedikt, Nicole Fraser-Hurt, Zara Shubber, Nejma Cheikh, Stela Bivol, Anna Roberts, David P. Wilson, Rowan Martin-Hughes.

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
