## [Decision Letter · Decision Letter 0]

15 Feb 2023

PGPH-D-22-01319

Public health benefits of shifting from inpatient to outpatient TB care in Eastern Europe: optimising TB investments in Belarus, the Republic of Moldova, and Romania

Dear Dr. Kelly,

Thank you for submitting your manuscript to PLOS Global Public Health. After careful consideration, we feel that it has merit but does not fully meet PLOS Global Public Health’s publication criteria as it currently stands. Therefore, we invite you to submit a revised version of the manuscript that addresses the points raised during the review process.

We look forward to receiving your revised manuscript.

Kind regards,

Alice Zwerling, PhD

Academic Editor

Journal Requirements:

1. In order to meet journal requirements for reporting and reproducibility, at this time we request that you please update the Methods to report the original source of the data and the methods used to collect it in sufficient detail for another researcher to access the same data. In your Methods section, please include additional information about your dataset. 

a. Please clarify all sources of funding (financial or material support) for your study. List the grants (with grant number) or organizations (with url) that supported your study, including funding received from your institution. 

b. State the initials, alongside each funding source, of each author to receive each grant.

c. State what role the funders took in the study. If the funders had no role in your study, please state: “The funders had no role in study design, data collection and analysis, decision to publish, or preparation of the manuscript.”

d. If any authors received a salary from any of your funders, please state which authors and which funders.

Additional Editor Comments (if provided):

Reviewers' comments:

Reviewer's Responses to Questions

**Comments to the Author**

1. Does this manuscript meet PLOS Global Public Health’s publication criteria? Is the manuscript technically sound, and do the data support the conclusions? The manuscript must describe methodologically and ethically rigorous research with conclusions that are appropriately drawn based on the data presented.

Reviewer #1: Partly

Reviewer #2: Yes

2. Has the statistical analysis been performed appropriately and rigorously?

Reviewer #1: Yes

Reviewer #2: Yes

3. Have the authors made all data underlying the findings in their manuscript fully available (please refer to the Data Availability Statement at the start of the manuscript PDF file)?

Reviewer #1: No

Reviewer #2: No

4. Is the manuscript presented in an intelligible fashion and written in standard English?

Reviewer #1: Yes

Reviewer #2: Yes

5. Review Comments to the Author

Reviewer #1: Comments to authors

Thank you for allowing me to review this manuscript, which reports on a modeling study of TB programs in Belarus, Moldova, and Romania. The authors demonstrate the benefit of transitioning from hospital-based TB care to ambulatory TB care. While the study design is interesting and yields valuable results, the additional benefit of this manuscript compared to the three World Bank country reports that were already published needs to be clarified. I also have additional comments detailed below which I hope are beneficial.

General

1. From my understanding, this article is essentially a synthesis of 3 World Bank reports from 2017, 2018, and 2019. These reports, however, seem to have different (cost) data bases and different terminology of TB treatment regimens. The authors should try to merge terminology of TB treatments whenever possible (to allow the reader to compare between the 3 countries) and need to be more precise with respect to the cost data basis and assumptions for each country analysis.

2. Additionally, it is not entirely clear to me what the additional benefit of this manuscript is compared to the three World Bank country analyses that were already published. The authors should state where they describe what is already published in those country reports and what the additional “original” content of the manuscript at hand is.

Abstract

1. The background section focuses on the epidemiology and costs of DR-TB. However, from my understanding, this does not directly relate to the study that the authors conducted, as the study’s focus is not DR-TB. The authors analyzed the distribution of resources between ambulatory and hospital-based DS-TB and DR-TB treatment. Hence, I would suggest that the authors motivate their study based on the question of ambulatory vs. hospital-based treatment, not on DR-TB treatment.

2. The findings section appears very brief (it is even shorter than the background section). It barely mentions concrete findings from the 3 countries. If word limit permits, I would suggest that the authors add more key findings to the abstract (e.g., concrete findings from the 3 countries).

Introduction

3. I found it hard to follow the reasoning of the introduction. In the first sentences, the authors mention that there has been a slow transition towards outpatient TB care in Eastern Europe. How does this causally relate to the high DR-TB rate in Eastern Europe mentioned in the second part of the first paragraph? Inpatient-centered TB care in FSU countries may have contributed to the rise of DR-TB, but it appears to be a multifactorial phenomenon. Thus, I suggest that the authors provide more info on inpatient care and rise of DR-TB in Eastern Europe/FSU countries in the intro (see, e.g., doi: 10.5588/ijtld.14.0190 or doi: 10.3855/jidc.3396). Additionally, I suggest you cite the most recent WHO global TB report.

4. I suggest you consider citing the systematic reviews by Fitzpatrick et al. 2012 (doi: 10.2165/11595340-000000000-00000) and Bassili et al. 2013 (doi: 10.4269/ajtmh.13-0004) when making the case for ambulatory care in the introduction.

5. The first and third paragraph of the introduction start with the same sentences, and it seems they make very similar arguments. Consider re-writing the intro to have a coherent line of reasoning instead of reiterating similar arguments multiple times.

6. In the fourth paragraph of the introduction, the authors describe a shift towards outpatient TB care during the Covid-19 pandemic. Please provide a reference for your claims.

7. The introduction focuses on DR-TB treatment, but DS-TB is also part of the modeling. I suggest the authors also include WHO recommendations for ambulatory DS-TB treatment in the introduction.

Methods

8. Table 2 summarizes the outpatient-based treatments for Belarus, Moldova, and Romania. Although I understand that there are differences in TB drug regimens and terminology between the countries, I struggle understanding the table: For instance, what does “Considered” mean compared to “standard and incentivized”? What is “incentivized”? For MDR-TB, what is “MDR classic” and “MDR plus”, and how does this compare to “MDR standard”? What do the authors understand as short-course MDR-TB treatment (with injectables or without injectables or the BPaL(M) regimen)? I think additional explanations and definitions on the terminology of drug regimens mentioned in the table would aid understanding. Additionally, the authors could standardize the terminology across countries (referring to the terminology of WHO-recommended treatments). This would help synthesize the separate analyses of the three countries.

9. The Outcomes section of the article stipulates that ambulatory treatment is superior compared to hospital-based treatment. This already constitutes results/interpretations instead of methodology. In the “Outcomes” paragraph, one would expect an explanation of the outcome variables of interest.

10. According to the methodology, the authors performed an optimization of TB care spending for each country, which was not limited to the question of investing in hospital-based or ambulatory care, but also included other treatment interventions (e.g., palliative care, prison-based treatment). Did the authors pre-define that an “optimized” scenario had to include mostly ambulatory care (and thereby pre-defined some parameters of the optimized solution), or did ambulatory care simply come up as the optimal solution for each country? This comment also touches the study motivation: Did the authors conduct a study to determine the potential benefits of ambulatory over hospital-based care, or did the authors conduct a study to optimize TB care in the respective countries? At the moment, these two distinct study motivations are mixed throughout the manuscript.

11. Most likely, despite shifts towards ambulatory care, some TB patients will still require hospitalization due to their severity of illness or comorbidities. How was this factored in the model?

12. As the reports for the 3 countries stem from different years, how did the authors standardize costs to the same year and currency? What was the currency year used in the analysis?

13. What was the perspective of the cost analyses in the 3 countries? Were costs determined from a societal perspective or health system perspective? What was the time horizon of cost collection? What was the discount rate? This basic data of economic evaluations needs to be included in the methods section.

Results

14. Minor: Please add a y-axis label in Figure 2.

15. Figure 2: What do the authors mean by “TB treatment coverage”? Is it the number of people treated for the respective TB type? And do the optimization bars refer to the year 2035? Do the optimized scenarios include re-investment of saved resources in TB care?

16. Figure 3: Do the projections of TB prevalence, active TB cases, and TB-related deaths until 2035 include a re-investment of resources that were freed up by re-allocation of resources?

17. General note: I suggest the authors revise the results to clearly reflect the time dimension of each of their results. I found it difficult to understand whether the authors discuss potential savings that can be achieved by redistribution at the initial year of analysis or savings projections until 2035, and whether these include a re-investment of freed resources in other TB interventions.

18. For some TB resistance types, Table 1 stipulates more than one treatment option. For example, DS-TB treatment in Belarus may be “standard” or “incentivized”, or long-course MDR-TB treatment in Moldova may be “MDR classic” or “MDR plus”. The results, however, only discuss the general distribution of resources among TB resistance types and ambulatory vs. hospital-based care. How were resources distributed between different treatment options within the same TB resistance type?

19. Consider adding subheadings (maybe one subheading for each country) to the results section.

Discussion

20. Meta-analyses (some of which you cite) indicated that decentralized care has similar or even better treatment outcomes than hospital-based care, but this should be taken with a grain of salt. Evidence stems from cohort studies (some with historical treatment groups), and there are no randomized trials at hand. Existing studies are therefore prone to selection bias, where sicker patients tend to receive hospital-based treatment (and tend to have worse outcomes), and patients who are not so ill are treated ambulatorily (with better outcomes). This argument is worth mentioning in the discussion.

21. As mentioned in a previous comment on the introduction, the authors need to provide references in the paragraph on Covid-19 to back up their statements on transitions to ambulatory care that took place during the Covid-19 pandemic.

22. On a similar note, is there evidence for the following 4 claims the authors make in the discussion? “Other benefits include reduced nosocomial transmission-related health systems costs, cost (direct and indirect) to the patient, as well as reduction in infection risk, and stigma surrounding access to longer-term hospital care.”

23. With results from the Nix, ZeNix, and TB PRACTECAL trials, recommended DR-TB treatment has been changing dramatically over the last 5 years. Changing DR-TB drug regimens and treatment durations appear to be a major limitation of findings of the modeling that need to be addressed in the discussion.

24. The concluding paragraph seems to be unrelated to the results of the study. Ideally, this paragraph should reflect key results of the study in conjunction with central interpretations/recommendations.

Reviewer #2: Dear authors,

Thank you for submitting the paper titled “Public health benefits of shifting from inpatient to outpatient TB care in Eastern Europe: optimising TB investments in Belarus, the Republic of Moldova, and Romania” for potential publication in PLOS Global Public Health as a Research Article. The topic is interesting, and the use of the relatively new tool Optima TB may reveal some new lessons about the adoption of outpatient DR TB care in the region. I believe it can be published if some minor revisions are made to the current version of the paper.

Introduction:

1. At four pages, including a table, I find the introduction to be unduly long. Authors should reduce this section by summarising more or moving parts of this to the discussion section.

2. Line 50 attributes the increases in rates of new cases of MDR/RR to the inpatient care system in place. Can they provide references to support this?

3. The last phrase in line 87 is a repetition of what was state in line 85.

4. The sentences starting in line 104 and ending line 107 can be moved to methods.

5. It is not exactly clear what ‘these studies’ in line 109 refer to – is it the current study or the cited studies?

6. Lines 109 to 124 could be moved to discussion.

Methods:

7. Author name and date missing in line 129 before (15) to complete the sentence.

8. Even though the authors reference in the (15) reference, a brief description of Optima TB will be helpful here – including how it was developed and validated and if it has been used successfully in similar contexts.

9. The sentence starting on line 130 on stakeholders is out of place in this section and should be moved to the ‘Study data and costing’.

Results:

10: I believe this section will be clearer if reorganised. I propose discussing each country separately before proceeding to how they compare. Although the authors have attempted to do that, there are sections discussing the 3 countries in between the texts for each country, which is a bit distracting. Eg, after beginning with Belarus, the authors mention Moldova and Romania in lines 250 to 251, then return to Belarus in lines 251 before going on to talk about Moldova in lines 264-280 and very briefly to Romania in 282 to 286. Authors could move the Moldova and Romania mention on line 250 to after line 286.

Discussions:

11. Can you discuss how your study compares to similar modeling studies in similar contexts?

12. Please add a paragraph on the limitations of your approach and findings.

6. PLOS authors have the option to publish the peer review history of their article (what does this mean?). If published, this will include your full peer review and any attached files.

**Do you want your identity to be public for this peer review?** For information about this choice, including consent withdrawal, please see our Privacy Policy.

Reviewer #1: **Yes: **Nicolas Paul

Reviewer #2: No

---

## [Decision Letter · Decision Letter 1]

15 May 2023

Public health benefits of shifting from hospital-focused to ambulatory TB care in Eastern Europe: optimising TB investments in Belarus, the Republic of Moldova, and Romania

PGPH-D-22-01319R1

Dear Dr Kelly,

We are pleased to inform you that your manuscript 'Public health benefits of shifting from hospital-focused to ambulatory TB care in Eastern Europe: optimising TB investments in Belarus, the Republic of Moldova, and Romania' has been provisionally accepted for publication in PLOS Global Public Health.

Best regards,

Alice Zwerling, PhD

Academic Editor

Reviewer Comments (if any, and for reference):

Reviewer's Responses to Questions

**Comments to the Author**

1. If the authors have adequately addressed your comments raised in a previous round of review and you feel that this manuscript is now acceptable for publication, you may indicate that here to bypass the “Comments to the Author” section, enter your conflict of interest statement in the “Confidential to Editor” section, and submit your "Accept" recommendation.

Reviewer #1: All comments have been addressed

Reviewer #2: All comments have been addressed

2. Does this manuscript meet PLOS Global Public Health’s publication criteria? Is the manuscript technically sound, and do the data support the conclusions? The manuscript must describe methodologically and ethically rigorous research with conclusions that are appropriately drawn based on the data presented.

Reviewer #1: Yes

Reviewer #2: Yes

3. Has the statistical analysis been performed appropriately and rigorously?

Reviewer #1: Yes

Reviewer #2: Yes

4. Have the authors made all data underlying the findings in their manuscript fully available (please refer to the Data Availability Statement at the start of the manuscript PDF file)?

Reviewer #1: Yes

Reviewer #2: Yes

5. Is the manuscript presented in an intelligible fashion and written in standard English?

Reviewer #1: Yes

Reviewer #2: Yes

6. Review Comments to the Author

Reviewer #1: Thank you for allowing me to review the revision of the manuscript. The authors comprehensively responded to my previous comments. I have nothing further to add. Congratulations.

Reviewer #2: Dear authors,

Thank you for revising the manuscript as recommended.

Best regards.

7. PLOS authors have the option to publish the peer review history of their article (what does this mean?). If published, this will include your full peer review and any attached files.

**Do you want your identity to be public for this peer review?** For information about this choice, including consent withdrawal, please see our Privacy Policy.

Reviewer #1: **Yes: **Nicolas Paul

Reviewer #2: No
